



# Slow strain waves in blocky geological media from GPS and seismological observations on the Amurian plate

**V.G. Bykov and S.V. Trofimenko**

Institute of Tectonics and Geophysics, Far Eastern Branch, Russian Academy of Sciences,
65, Kim Yu Chen St., Khabarovsk, 680000, Russia
*Correspondence to*: V.G. Bykov (bykov@itig.as.khb.ru)

**Abstract.** Based on the statistical analysis of spatiotemporal distribution of earthquake epicenters and perennial geodetic observation series, new evidence is obtained for the existence of slow strain waves in the Earth. The results of our investigation allow us to identify the dynamics of seismicity along the northern boundary of the Amurian plate as a wave process. Migration of epicenters of weak earthquakes ($2 \leq M \leq 4$) is initiated by the east-west propagation of a strain wave front at an average velocity of 2.7 km/day. We have found a synchronous quasi-periodic variation of seismicity in equally spaced clusters with spatial periods of 3.5° and 7.26°comparable with the length of slow strain waves. The geodetic observations at GPS sites in proximity to local active faults show that in a number of cases, the GPS site coordinate seasonal variations exhibit a significant phase shift, whereas the time series of these GPS sites differ significantly from a sinusoid. Based on experimental observation data and the developed model of crustal block movement we have shown that there is one possible interpretation for this fact that the trajectory of GPS station position disturbance is induced by migrating of crustal deformation in the form of slow waves.

**Key words:** background seismicity, seismic clusters, strain waves, Amurian plate, space-time seismicity model, oscillatory movements of crustal blocks.

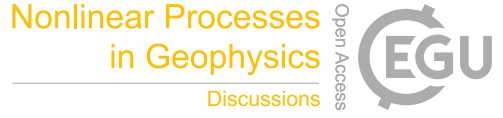

## 1 Introduction

The inhomogeneous blocky structure of the crust and the lithosphere considerably affects the deformation, seismic, filtration and other processes. The effect of the blocky structure on the distribution of earthquakes can be especially clearly traced. It is exactly the blocky structure of the geological medium which results in the generation of waves of different types including slow strain waves (Bykov, 2008). Clarification of the link between movements of tectonic structures and slow strain wave processes is of fundamental importance for expanding our understanding of the physics of earthquakes.

The most important problem of recent geodynamics is to clarify the mechanisms responsible for the propagation of the energy of deformation processes and tectonic stress transfer at the boundaries between the blocks and the lithospheric plates, and to explore the causes of migration of earthquake epicenters. The problem has been argued for more than 45 years since Elsasser's publication (1969), suggesting the equation of local stress transfer in the rigid elastic lithosphere underlain by the viscous asthenosphere. The possibility of using Elsasser's model to describe migration of seismicity was further discussed in papers published by other researchers. Bott and Dean (1973) introduced the term "stress or strain waves" and obtained the expression for the velocity of the wave propagating along the lithospheric plate. According to their calculation, the stress wave velocity attains to 0.1-100 km/yr. Anderson (1975) generalized Elsasser's model in order to elucidate the mechanism of earthquake migration in the subduction zone and estimated the stress wave velocity along the island arc about 50-170 km/yr. In the model developed by Ida (1974), the solution was obtained in the shape of "slow-moving deformation pulses" propagating along the fault at a constant velocity. The gouge viscosity and thickness variations in the fault yield the pulse velocity ranging from 10-100 km/yr to 1-10 km/day. The first interval corresponds to earthquake migration velocities at a wavelength of about tens of kilometers, whereas the second interval is compliant with aseismic creep at about 1 km wavelength. Scholz (1977) introduced the concept of the "deformation front" to describe large-scale tectonic processes triggering large earthquakes. As estimated by Scholz, the velocity of the deformation front propagating through NE China, that triggered the 1975, M=7.3 Haicheng earthquake, attained to 110 km/yr.

The advances in theoretical studies of slow strain waves in the Earth initiated the search for the possibilities of experimentally detecting the propagation effects of the waves of this type, and, in the first place, the intense study of earthquake migration. By now, the deformographic, geodetic and hydrological measurements performed worldwide have revealed the migration of deformations at velocities of about 10-100 km/yr and 1-10 km/day (Kasahara, 1979; Bella et al., 1990; Harada et al., 2003; Kuz'min, 2012; Reuveni et al., 2014; Yoshioka et al., 2015).



Migration of earthquake epicenters coincides with the velocity (10-100 km/yr) and direction of
crustal deformation movement (Kasahara, 1979; Barabanov et al., 1988) and with hydrological
effects (Kissin, 2008). Furthermore, the absorption and dispersion of the waveforms of
migrating deformation were detected (Kasahara, 1979; Barabanov et al., 1988), i.e., the main
properties of a common wave process. In terms of the physical mechanism of propagation, slow
strain waves are similar to common seismic waves, but the fundamental difference is that they
propagate at super low velocities, ultra low frequencies and have large wavelength (Bykov,
2005). This hampers the direct instrumental measurements of strain waves and the concomitant
effects.
In the present study, we have obtained new evidence of the existence of strain waves in the
Earth resting upon a comprehensive statistical analysis of the dynamics of seismicity along the
northern boundary of the Amurian plate and the data derived from in situ GPS experimental
observations performed near this boundary.

**2  Methods for detection of slow strain waves**

Slow strain wave transmittance through the fault-blocky geological medium is
accompanied by various seismic, hydrogeological, electrokinetic, geochemical and other effects.
The methods for strain wave detection are divided into indirect, that display the wave-shaped
variations in the geophysical fields due to temporal variations of the stress state of the medium,
and direct ones immediately recording the migration of deformations.
The seismic, geoelectric and geochemical methods of strain wave recording are referred to
the indirect methods. Indirect evidence of the existence of strain waves is manifested in the
targeted migration of large earthquakes (Stein et al., 1997), the occurrence of seismic velocity
anomalies (Lukk and Nersesov, 1982; Nevskii et al., 1987), a cyclic wandering of aseismic
strips in the Earth's mantle (Malamud and Nikolaevsky, 1983; 1985); oscillatory movements of
the seismic reflection sites (Bazavluk and Yudakhin, 1993; Bormotov and Bykov, 1999) and the
migration of geophysical field anomalies (radon, electrokinetic signals) in proximity to faults
(Nikolaevskiy, 1998).
The direct indications of strain waves are displayed in wave fluctuations of the ground
water level and the migration of slopes and surface deformations. The direct methods exploring
temporal variations of crustal deformation comprise the deformographic (Kasahara, 1979; Ishii
et al., 1983; Nevskii et al., 1987; Bella et al., 1990; Harada et al., 2003), hydrogeodynamic
(Barabanov et al., 1988; Kissin, 2008) and geodetic measurements (Kuz'min, 2012) including
the methods of deformation measurements using laser ranging (Milyukov et al., 2013) and GPS
observations (Reuveni et al., 2014; Yoshioka et al., 2015).





To detect the main physical mechanisms of seismicity migration and the generation of
signals of different nature that are accompanying strain waves, we need performing further
observations and improving GPS- and seismological data processing technique, and conducting
theoretically prepared and purposeful experiments.
The answer to the question "where to search for slow strain waves?" is directly linked
with the detection of the main types of tectonic structures generating these waves.
**3   Tectonic structures generating slow strain waves**

From the published results it follows that subduction, collision, active riftogenesis and
transform fault zones are the most probable types of tectonic structures generating strain waves.
These intensive sources of different tectonic nature possess a common property − they are the
interaction zones between crustal blocks and the lithospheric plates.
Migration of shear deformation in subduction zones is directed from the ocean toward the
coast. This general tendency was first revealed in area of the Japan island arc where migration is
oriented east-west, and in the opposite Pacific coastal area − in the western Cordilleras, where
deformations migrated from south to north (Kasahara, 1979). Migration of the maximum of the
vertical crustal deformation from the subduction zone toward the continent at a velocity of
about 10 km/yr was also observed near the Tohoku region (northeastern Japan) and the Izu
Peninsula (central Japan), where the Pacific and Philippine plates subduct beneath the Eurasian
plate (Miura et al., 1989). All these data reasonably lead to an assumption that subduction zones
are one of the possible sources of slow strain waves.
The seismicity pattern observed in the south of Middle Asia can also be explained by strain
waves excited under the oscillating regime of the Eurasian and Indian lithospheric plate collision
in the Pamir and Tien Shan junction zone (Nersesov et al., 1990). The compression at the
Indostan and Eurasian lithospheric plate boundary in the Himalayan collision zone is the source
of "fast" and "slow" waves of plastic deformation that trigger earthquakes in Central and East
Asia (Wang and Zhang, 2005).
In the Baikal rift system, four main groups of strain waves with different velocities (7-95
km/yr) and lengths (130-2000 km) are distinguished that cause recent activation of seismoactive
faults in Central Asia (Gorbunova and Sherman, 2012).
Based on continuous long-term seismic and laser ranging observation data, it has been
established the effect of propagation of slow waves of tectonic deformations traveling along
transform faults at velocities of 40-50 km/yr at the lithospheric plate boundaries in Southern
California and the Kopet-Dag region (Nevskii et al., 1987). Seismicity variations along the



Pacific and North American plate boundary in the San-Andreas transform fault zone (California) are also suggested to be associated with "slowly traveling strain waves" (Press and Allen, 1995).

The rotational block movements in the fault zones due to tectonic processes or earthquakes are considered one of the main physical mechanisms of strain wave generation (Nikolaevskiy, 1996; Lee et al., 2009; Teisseyre et al., 2006).

## 4  Seismic effects of slow strain waves at the northern edge of the Amurian plate

In order to specifically investigate the relationship between strain waves and the dynamics and seismicity pattern observed in fault-blocky geological media, we have selected the study area on the northern margin of the Amurian plate – the most seismically active area of the interaction zone between the Amurian and Eurasian plates.

The analysis of the spatiotemporal seismicity pattern observed in vast regions is commonly performed based on statistical processing of earthquake catalogues. The directions of earthquake epicenter (or groups of epicenters) displacements are defined and their displacement rates are determined. As opposed to the standard regional approach, we here applied a comprehensive analysis including both conventional statistical methods and those of cluster analysis adapted by the authors for the geodynamic zone gradation. The details of developed clustering technique and statistical analysis of background seismicity can be found in (Trofimenko et al., 2015).

To study the dynamics of seismicity in different zones, the area along the northern boundary of the Amurian plate was divided into separate clusters (Fig. 1). When clustering, we applied the criterion of earthquake grouping near active faults, and the geomorphological and tectonic features of active structures, as well as the presence of meridional (submeridional) first-rank faults within the distinguished zones, were taken into consideration.

When developing space-time models of seismicity, the spatial relationship between separate seismic clusters during a year was revealed and taken into account. Based on statistical distributions of earthquakes, the analysis of seismicity maxima passage over east-westerly arranged clusters has been performed.

The basic data were derived from the catalogue "Earthquakes of Russia" (http://eqru.gsras.ru), the catalogue compiled by the Baikal Branch of the Geophysical Survey of the Russian Academy of Sciences (GS RAS) (http://www.seis-bykl.ru/) and the IRIS catalogue (http://www.iris.edu).

As a result of the calculation, the average period of seismicity maximum passage in days from the beginning of the year has been determined for each cluster, which is assigned to the average value of the cluster longitude. These values were used for the calculation of the



displacement rate of seismicity maxima. We calculated the velocities and wavelengths of slow
strain waves from the maxima of the spatial correlation of seismicity.
The spatiotemporal distributions of earthquake epicenters reflect synchronization of
seismicity maxima in the annual cycles over a certain spatial interval (migration period). The
statistical calculations performed for each cluster allowed the identification of six similar
spatiotemporal cycles of seismicity maxima migration A, B, C, D, E and F (Fig. 1), for which the
spatial periods of migration and displacement rates of seismicity maxima have been calculated.
In the northeastern segment, the maxima of statistical distributions are located in the
clusters arranged nearly equally apart from each other, at $L_{A-C} = (7.26 \pm 0.74)°$, which corresponds
to a distance of 360-420 km for a range of investigated latitudes. For the northwestern segment,
the spatial period is equal to $L_{D-F} = (3.8 \pm 0.5)°$, at the average, which corresponds to half of the
interval $L_{A-C}$, or a distance of 210-270 km (Fig. 1). In the study area, the parameter $L_{A-C}$ is equal
to double the distance between the main structural-tectonic elements of the Earth's crust and
corresponds to double the size of tectonic inhomogeneities revealed from the geophysical field
anomalies (Trofimenko, 2010).
The determined spatial period $L_{A-C} = 7.26°$ (360-420 km) is comparable with the
wavelength $\lambda = 250\text{-}450$ km of slow strain waves observed in the study area on the northern
margin of the Amurian plate (Pribaikalya and Priamurye areas lying within 107°E-140°E)
(Sherman et al., 2011). The direction of the seismicity maxima displacement coincides with the
displacement vector of the strain wave front (Sherman, 2013) (Fig. 2).
The displacement rate values for seismicity maxima are obtained from regression
equations using the linear approximation method and are equal to $U_A = -2.6$ km/day, $U_B = -3.2$
km/day, $U_C = -2.7$ km/day, $U_D = -2.61$ km/day, $U_E = -2.83$ km/day and $U_F = -2.15$ km/day for spatial
cycles A, B, C, D, E and F, respectively. The minus sign means the westward displacement of
the seismicity maxima.
For the entire northeastern segment, the average value of the velocity modulus of the
seismicity maxima displacement (with a relative determination error of 7%) is equal to $U_{A-C} =$
2.75-2.80 km/day, whereas for the northwestern segment this value is $U_{D-F} \approx (2.5 \pm 0.3)$ km/day.
The seismicity maxima displacement rate value is $U_{A-F} \approx (2.68 \pm 0.34)$ km/day or about 1000
km/yr along the entire northern boundary of the Amurian plate.

## 5  The slow strain wave effects inferred from GPS observations


To explore the deformation processes in the geological medium with a discrete blocky
structure and to perform special GPS experimental observations, we selected the South Yakutia
geodynamic polygon located near the northern boundary of the Amurian plate, at the junction of



two major tectonic structures – the Aldan Shield and the Stanovoy Range. Recently, a number of
blocks of different size and configuration have been inferred here from geological data. These
blocks experience the vertical and horizontal movements of different directions, velocities and
amplitudes (Imaeva et al., 2012), which are responsible for a complicated character of tectonic
movements.
We have analyzed a set of time series obtained at two of collocated GPS sites NRGR and
NRG2 situated near the active fault intersection area in the central part of the Stanovoy Range
(Fig. 1). The NRGR site is located in area of the Chulman depression on $15 \times 20$ km$^2$ size
microblock and is involved in different types of crustal movements and deformations in
consistency with the kinematics of the bordering active faults. The site NRG2 location is
approximately 2 km south of the NRGR site and closer to the zone of influence of the active
Berkakit fault. The GPS time series obtained at stations NRGR and NRG2 for the horizontal and
vertical components are shown in Fig. 3. The stable long-period displacement component is
typical for both observation sites in the southeastern direction. For the vertical and horizontal
components observed in other directions, the course of the annual displacements is absolutely
different. At the two observation sites, the horizontal displacement components in the "North-
South" direction are represented by in-phase curves that can be approximated by a sinusoid (Fig.
3 a). The vertical and horizontal displacement components in the "East-West" direction vary in
an anti-phase manner during separate periods of measurements (Fig. 3 b, c), which contradicts
the common dynamics of long-period components. The shapes of these curves for the horizontal
displacement components are appreciably different from a sinusoid.
It is necessary to emphasize that the meteorological factors in the annual cycles influence
the shapes of the movement trajectories of the collocated sites equally (van Dam et al., 1994).
Therefore, the detected paradox cannot be explained by the meteorological causes.
This paradox can only be resolved in the case when the observation sites are adjacent to the
boundaries of specific – "hinge" – type local faults (Fig. 4 a). Really, for the site NRGR, a local
feathering fault of the Sunnangyn-Larba northeast-trending fault system is the "hinge", whereas
for the site NRG2, the" hinge" is one of the branches of the Berkakit northwest-trending fault
(Fig. 1). The physical model of this fault-blocky structure can be represented as a set of rods –
physical pendulums (Fig. 4b), whose lower parts are fixed, while the upper parts are disturbed
from the equilibrium condition. In this case, the upper parts of the rods (blocks) are displaced
with respect to some central line (the fault hinge).
The approximation curve fitting for the vertical component of block displacement has led
to one more unexpected result. The shape of the best fit function approximating the experimental
curve appeared to coincide with a breather – the solution (2) of the sine-Gordon equation (see




below). When selecting the theoretical curve in the shape of a breather (2), this result for the
"North-South" component is obtained at $\omega = 0.873$ with an error equal to 0.048 (for the sine
0.069),while for the "East-West" component − at $\omega = 0.780$ with an error equal to 0.052 (for the
cosine 0.149). The approximation error of experimental data is calculated from the formula
$\sigma = \sqrt{1/12 \sum_{k=1}^{12} (Y_k^E - Y_k^T)^2}$ , where $Y_k^E - Y_k^T$ are the residuals between the observed and calculated
monthly averaged station positions for the sinusoid and breather. The shapes of the fitted curves
are shown in Fig. 5.
The coincidence of the trajectory shape of measured vertical displacements with the shape
of a breather, and the correspondence of the blocky structure in area of GPS site locations to the
model of coupled pendulums served as a motivation for application of the sine-Gordon equation
to describe the evolution of the vertical components of block movements.
The mathematical model of quasi-periodical vertical components of oscillations of rigidly
coupled crustal blocks with the adjacent "hinge"-type faults corresponds to the equation:

$$\frac{\partial^2 \varphi}{\partial \eta^2} - \frac{\partial^2 \varphi}{\partial \xi^2} = \sin \varphi , \qquad (1)$$

$\eta = \omega t , \; \xi = x\omega/c , \; \omega^2 = mgl/I , \; c^2 = \tau d^2/I ,$

where $\varphi$ is the angle of deviation of the pendulum (rod) from the equilibrium position;
$mgl \sin \varphi$ is the moment of the gravity force, $m$ is the lamped mass of the pendulum, $l$ is the
length of the rod (the height of the block), $\tau d^2 \dfrac{\partial^2 \varphi}{\partial x^2}$ is the sum of the moments of the torsion
forces exerted by the adjacent blocks, $\tau$ is the constant of the spring torsion (rigidity), $d$ is the
increment of the interblock distance (increase or decrease depending on the type of movement),
$I$ is the moment of the block inertia.
One of the solutions of equation (1) is called a breather (dynamic soliton) and represents
a nonlinear function which, for the case of the soliton with the immobile center of gravity can be
written as:
$$\varphi(x,t) = 4\,\text{arctg}\left[ \left( \frac{\sqrt{1-\omega^2}}{\omega} \right) \frac{\sin(\omega t)}{\text{ch}(x\sqrt{1-\omega^2})} \right], \qquad (2)$$

where $\omega$ is the inner frequency of the breather, $x$ determines the origin of the curve and $t$ is the
independent variable (time).
Like a soliton, the breather has the shape of an impulse; it is localized in space and is
pulsating in time. In the low frequency range $\omega \ll 1$ the breather can be qualitatively treated as



a weakly coupled kink-antikink pair (the sine-Gordon equation solutions of opposite signs in the
shape of a topological soliton – a wave with a changeless profile in the shape of a kink) (Braun
and Kivshar, 2004).

299   The detected high correlation of the observed site displacement trajectories with the

theoretical curve corresponding to a breather allows us to suggest that the mechanism of these
oscillations can be associated with the occurrence of strain waves in the fault intersection system.
In this case, these waves can be qualitatively treated as standing waves of compression-extension
in the blocky geological medium.

304   The sine-Gordon equation solution in the shape of a breather has previously been applied

for modeling the wave dynamics of faults and strain waves (Mikhailov and Nikolaevskiy, 2000;
Gershenzon et al., 2009; Erickson et al., 2011). Mikhailov and Nikolaevskiy (2000) considered a
scenario when collision of two tectonic waves (kink-antikink collision) resulted in the
occurrence of large earthquake. The solution in the shape of a breather has also been applied for
the interpretation of the features of fault dynamics observed after the 1989 Loma-Prieta
earthquake (Gershenzon et al., 2009). Based on a modified Burridge–Knopoff model, the
solution has been obtained that corresponds to a localized failure – a breather that propagates
along a fault and is damping in the fault segment of the final length (Erickson et al., 2011). Wu
and Chen (1998) have earlier reduced a one-dimensional Burridge–Knopoff spring-block model
to the sine-Gordon equation and applied its solution in the shape of a solitary wave (kink) to
investigate earthquakes.

**6 Concluding remarks**

319   The accumulated facts indicate to the propagation of slow wave-like movements within the

crust and the lithosphere at different velocities on global and regional scales (Bykov, 2014). The
results of our investigation (the periodicity of the seismic components, spatial cycles with phase
shift of seismicity maxima, migration velocity of earthquake epicenters) and their comparison
with the known data allow us to identify the dynamics of seismicity along the northern boundary
of the Amurian plate as a wave process. We have revealed synchronous quasi-periodic seismicity
variations in equally spaced clusters with spatial periods of 7.26° and 3.5°, that are comparable
with the length of slow strain waves ($\lambda$=250-450 km), detected in the Eurasian and Amurian
tectonic plate interaction area (107°E-140°E) (Sherman, 2013). The slow strain wave velocity in
Pribaikalya and Priamurye attains to 5-20 km/yr and is comparable with the migration velocity
of crustal deformations (10-100 km/yr) from the Japan-Kuril-Kamchatka subduction zone (Ishii
et al., 1978; Kasahara, 1979; Yoshioka et al., 2015).



The calculated average displacement rate value of the maxima of weak seismicity (2≤M≤4)
along the northern boundary of the Amurian plate is about 2.7 km/day (~1000 km/yr), which is
two orders of magnitude larger than the velocity of slow strain waves (~10-100 km/yr). This may
imply that slow strain waves modulate variations of weak seismicity (2≤M≤4) during the year.
The displacement of seismicity in the annual cycles occurs from east to west and coincides
with the direction of migration of large earthquakes, strain wave fronts and crustal deformation
detected from direct deformographic and GPS measurements (Kasahara, 1979; Bella et al., 1990;
Harada et al., 2003; Yoshioka et al., 2015). The slow strain wave fronts are triggers of large
earthquakes (M>6) in the submeridional faults of the Amurian plate.
The spatial correlation of migration of seismicity and deformations as well as the migration
of deformations – two different manifestations of the geodynamic process – may mean that
seismicity migration is associated with the propagation of tectonic stresses in the form of slow
strain waves that cause a complementary load and subsequent earthquake occurrence. The
numerous results of observations of seismicity migration are hard to explain by other causes
rather than wave-like variations of the global and local stress fields.
The conclusions on the wave pattern of the deformation process are consistent with the
results of special experimental observations performed to explore crustal block interaction. The
seasonal course of displacements of GPS stations NRGR and NRG2, involved in the in situ
experimental observations, or of the deformations of the blocky structure of the crust, exhibits a
wave-like rather than a linear pattern. The wave-like displacements can be explained by
transmittance of slow strain waves.
Resting upon the statistical modeling, we have established the in-phase and anti-phase
changes of the components of the full displacement vector, the relative time delay of the maxima
and minima for separate components, and dissimilarity of the displacement trajectory from a
sinusoid. In order to describe the evolution of oscillations of the interacting blocks, a simple
mathematical model is proposed from which it follows the explanation of the observed specific
behavior of these blocks.
Based on experimental observation data and the developed model of crustal block
movement, we have shown that there is one possible interpretation for this fact that the trajectory
of GPS station position disturbance is induced by migrating of crustal deformation in the form of
slow waves.
*Acknowledgements.* The reported study was funded by RFBR according to the research project
No. 16-05-00097 a.



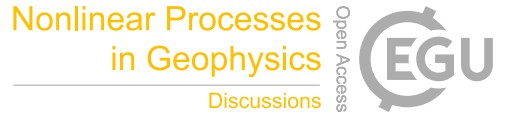

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



**Figures**

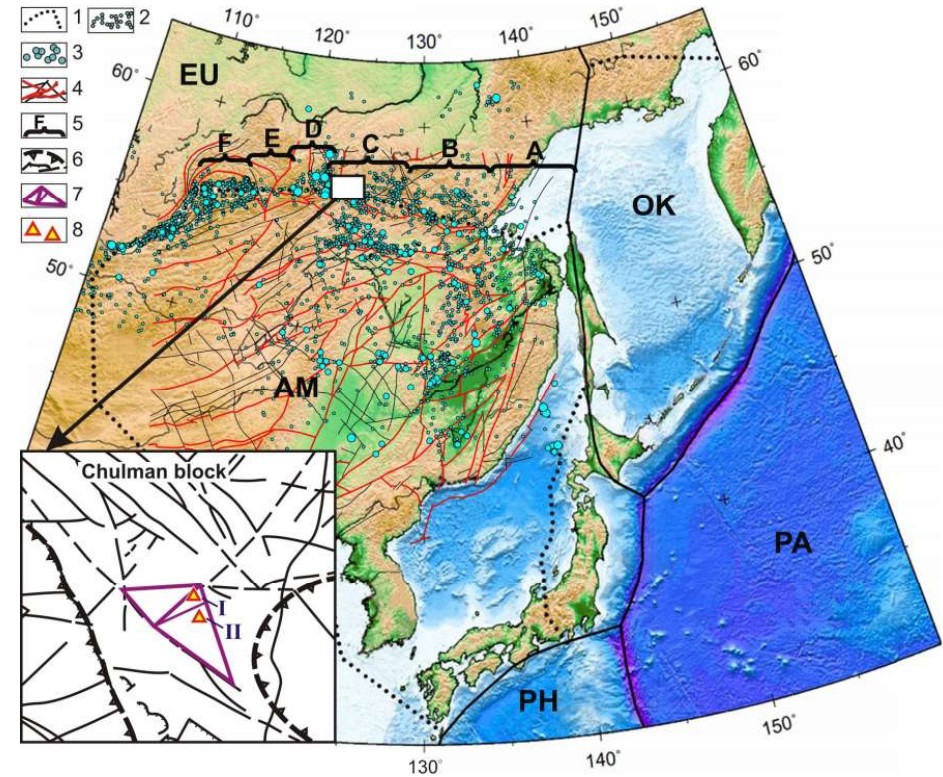


**Fig. 1.** The distribution of earthquake epicenters in the zone of interaction between the Amurian,
Eurasian and Okhotsk lithospheric plates.
1 – lithospheric plate boundaries: EU - Eurasian PA - Pacific, PH - Philippine, OK – Okhotsk; 2
– epicenters of earthquakes with magnitude M>3; 3 - epicenters of earthquakes with magnitude
M>5; 4 – main tectonic faulting; 5 – spatial cycles of seismicity.
A black rectangle shows a sketch map of fault tectonics of the Chulman block, where GPS sites
are located: 6 – northeast- and northwest-trending faults of different kinematics; 7 – local block,
bordered by active faults; 8 – GPS sites (I – NRGR, II – NRG2).



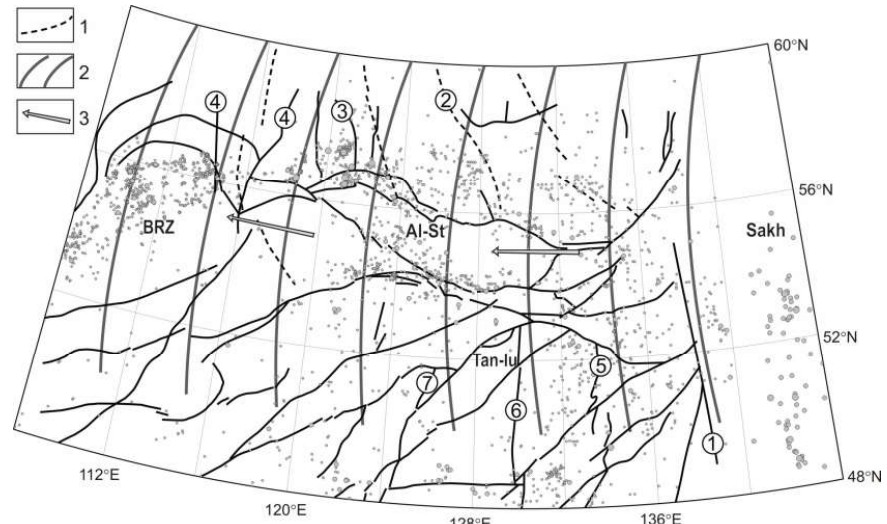

**Fig. 2** The spatial distribution of seismicity in the annual cycles with respect to the strain wave
fronts and meridional structures.
Active tectonic faulting: Tan-Lu fault zone, Aldan-Stanovoy block (Al-St) and Baikal rift zone
(BRZ). Figures in the circles denote the faults: 1 - Limurchan, 2 - Tyrkanda, 3 – Temulyakit
meridional faults, 4 – meridional structures of the eastern flank of the Baikal rift zone, 5 –
Gastakh, 6 – West-Turanian, 7 – Levo-Minsky.
1 – submeridional interblock faults of the Aldan shield; 2 – strain wave fronts (Sherman, 2013);
3 – the direction of seismicity maxima migration in the annual cycles and movements of the
strain wave fronts.














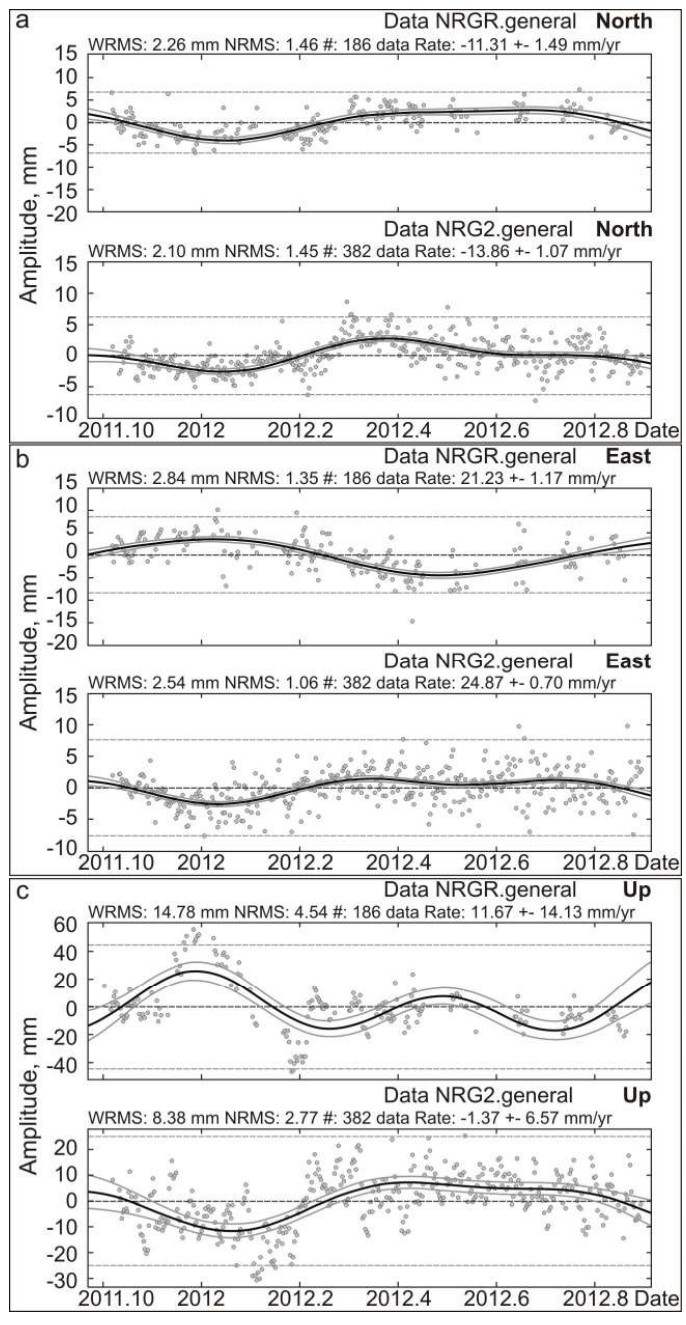


**Fig. 3** The dynamics of displacement components of NRGR and NRG2 station daily positions in
different directions.
a – for the N–S components; b – for the E–W components; c – for the vertical (Up–Down)
components.










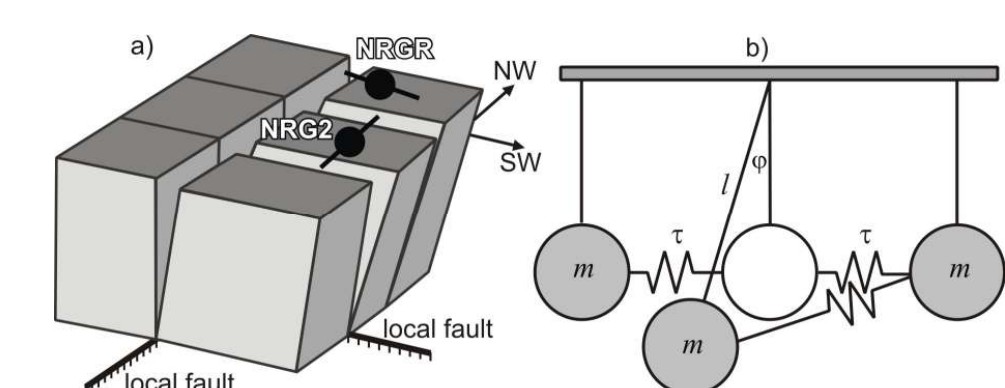

**Fig. 4** The generalized model of block movement in the vertical plane along differently oriented
local faults of the hinge type due to variable vertical loading. (a) The model of block movement
along NE- and NW-trending faults and schemes of the full displacement vector decomposition
into components. (b) The model of block movement in the shape of coupled pendulums
(notations are given in the text).




















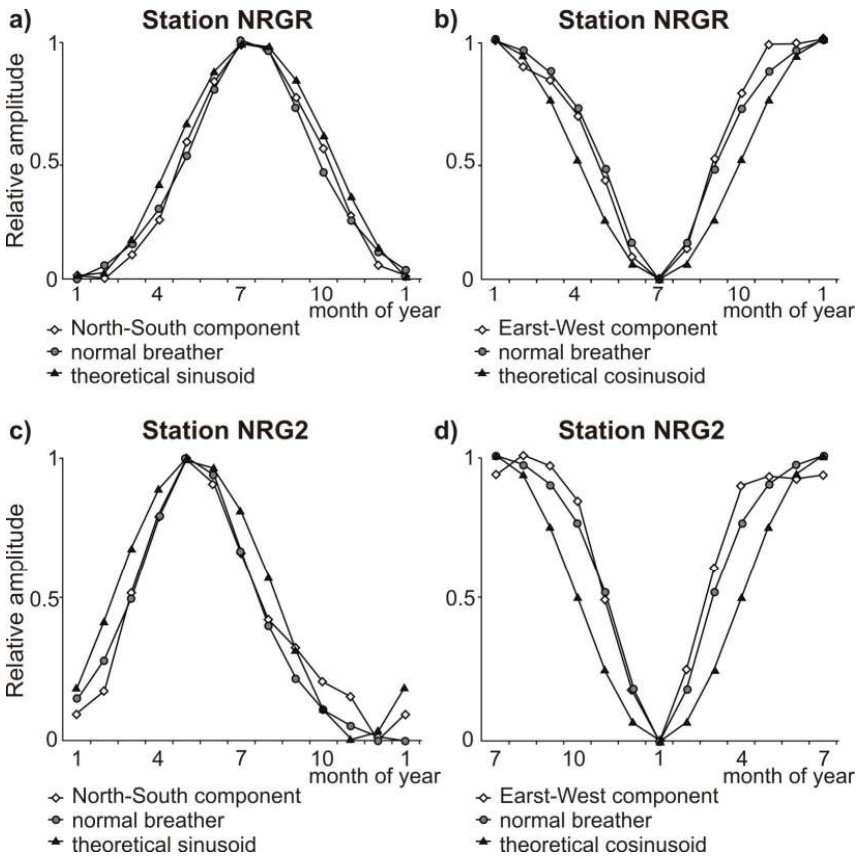


**Fig. 5** Seasonal variations of NRGR and NRG2 station positions.
Approximation of the observed displacement curves by the theoretical curves for the N–S (a) and
E–W (b) components at the NRGR site; for the N–S (c) and E–W (d) components at the NRG2
site.

601