# Peer review of "Slow strain waves in blocky geological media from GPS and seismological observations on the Amurian plate"

_Nonlinear Processes in Geophysics, 2016_

## Referee Comment (RC1) · S. Sherman (Referee) · 27 Oct 2016

I will not change my quick positive responses to the questionnaire on the review of the article and add a short note about it. Peer-reviewed article is devoted to one of the controversial issues of seismology and strain (deformation) waves. They are not diagnosed by instrumental methods due to the large wavelengths and very small velocities, their presence, and the parameters are estimated by indirect methods. The theoretical proof of the deformation waves is well described in the reviewed article, and shows their parameters. Shown practical importance to seismology – using the waves as a trigger mechanism, violate the delicate balance in fault-block environment of the lithosphere with subsequent movements in the fault line between which is generated

the earthquake. Most often such arguments are used in the analysis of major faults controlling the strong (M≥5.5-6.0) earthquakes. The originality and significance of the article is determined by the following achievements. The paper first waves as a trigger mechanism used for the region Amurian plate. The author analysis of deformation wave and their parameters based on the earthquakes of medium strength and weak. Often used of a strong earthquake. The author's methodology is well justified. Additionally I want to emphasize the high importance of the article, considering the time series of the GPS observations for horizontal and vertical component of modern movements. The authors demonstrated high correlation between observed trajectories of displacements of GPS points with the theoretical curve corresponding to the breather were a singular solitary wave. This suggests the occurrence of deformation waves in the system of intersecting faults. These waves authors interpretiruya as standing "wave of compression-extension" in the block medium of the lithosphere. The paper first showed that the rate of slow deformation waves in the territory of the Baikal region and the Amur region is 5-20 km/yr and in the order of magnitude is comparable with the rate of migration of crustal deformation (10-100 km/yr) from the Japan-Kuril-Kamchatka subduction zone (Ishii et al., 1978; Kasahara, 1979; Yoshioka et al., 2015). Calculated the average rate of displacement of the maxima of weak seismic activity (2≤M≤4) along the Northern boundary of the Amur plate has a magnitude of 2.7 km/day (∼1000 km/yr), which is one to two orders of magnitude greater than the speed of slow deformation waves (ïA¿10-100 km/yr). This may mean that the slow deformation waves modulate the change of weak seismic activity (2≤M≤4) during the year. The established correlation migration of seismicity and deformation may indicate that the first is a consequence of the movement of tectonic stress fields in the form of slow deformation waves, causing additional burden on the successive occurrence of earthquakes. Numerous observations of migration of seismicity is difficult to explain any other reasons besides the wave-like changes in the global and local stress field. This is an original and valuable conclusion. Article V. G. Bykov and S. V. Trofimenko "Slow strain waves in blocky geological media from GPS and seismological observations on the Amurian

plate" includes a new, largely original data on the activation of seismicity with a slow deformation waves in the Amur plate and recommended for publishing.

---

## Referee Comment (RC2) · Anonymous Referee #2 · 8 Nov 2016

Review of manuscript: Slow strain waves in blocky geological media from GPS and seismological observations on the Amurian plate V.G. Bykov and S.V. Trofimenko submitted for publication in "Nonlinear processes in geophysics"

General comments

The paper describes a study of earthquake epicenter migrations and GPS coordinate variations in application to the problem of slow strain waves in the lithosphere. As the data, the authors used the catalog of seismic events around the northern boundary of the Amurian plate and GPS data in the two observation points. It was obtained that the average displacement rate of the weak seismicity maxima in the studied region is about 2.7 km/day (∼1000 km/yr), which is significantly higher than the velocity of slow strain

waves (10-100 km/yr) reported in other papers. The main novel part of the article (in comparison with the previous publications of the authors) is devoted to analysis of GPS coordinate variations in time. A model based on sine-Gordon equation is suggested for explanation of the GPS data analysis results.

Specific comments and questions

1. The part 4 (Seismic effects of slow strain waves...) looks as a brief description of the previous papers of the same authors. I suggest to add a reference to the authors' paper published in Journal of Seismology, most of arising questions were clarified there. 2. The method used by authors to estimate the rate of earthquake epicenter migrations is not suitable for estimation of the values of displacements per day, it is far beyond the approximation accuracy. So, I suppose that authors should use values in km/year, not km/day. 3. It is not clear from the article text, how the graphs in Fig.5 were obtained from the data shown in Fig.3? Why the line in Fig.3a is called "sinusoid" while the line in Fig.3c is treated as "appreciably different from a sinusoid"? 4. The "pendulum" model suggested in the paper looks very artificial. It is unclear from the text, if the authors made the calculation for such a "pendulum" oscillations of tectonic blocks or just took one of the solutions of the sine-Gordon equation. If the calculations were made – what parameters were used?

Conclusion

I recommend to accept the paper for publication with minor corrections in accordance with the above comments.

---

## Author Comment (AC3) · 23 Nov 2016

Dear Professor S.I. Sherman,

We would like to thank you for your encouraging positive evaluation of our manuscript. We are also deeply grateful for your careful and meticulous reading of the paper and concrete and insightful comments and deep analytical conclusions on the manuscript which we really highly appreciate. Thank you once again for evaluating our contribution to the solution of the problem of wave geodynamic processes.

V. Bykov and S. Trofimenko

---

## Author Response (AR1)

Dear Editor Prof. Arcady Dyskin,

We are very grateful to you and the two reviewers for encouraging positive evaluation of the manuscript of our paper and constructive and thoughtful comments and suggestions, and also for the recommendations and corrections proposed by the anonymous reviewer to improve the quality of the manuscript. We have addressed all the issues raised in his review and have modified the text of the manuscript accordingly. The authors would like to kindly acknowledge both detailed reviews. Below are responses to the two reviewers and a summary of the changes performed following the anonymous reviewer's comments and recommendations.

Authors' response to Reviewer 1.

We would like to thank Prof. S. Sherman for his encouraging positive evaluation of our manuscript. We are deeply grateful for his careful and meticulous reading of the paper and concrete and insightful comments and deep analytical conclusions on the manuscript which we really highly appreciate. Thank you once again for evaluating our contribution to the solution of the problem of wave geodynamic processes.

Authors' response to Reviewer 2.

**The authors' answers to the second reviewer's comments:**

1. The part 4 (Seismic effects of slow strain waves. . .) looks as a brief description of the previous papers of the same authors. I suggest to add a reference to the authors' paper published in Journal of Seismology, most of arising questions were clarified there.

   We complemented the $2^{nd}$ paragraph of section 4 by a brief explanation and the reference to the authors' paper: Trofimenko, S.V., Bykov, V.G., and Merkulova, T.V.: Space-time model for migration of weak earthquakes along the northern boundary of the Amurian microplate, Journal of Seismology, 2016a. doi:10.1007/s10950-016-9600-x .
   The added sentence, highlighted in blue, is put in the text on lines 177, 178 and 179 of the revised manuscript.

2. The method used by authors to estimate the rate of earthquake epicenter migrations is not suitable for estimation of the values of displacements per day, it is far beyond the approximation accuracy. So, I suppose that authors should use values in km/year, not km/day.

   Taking into consideration that in the overview section of the manuscript all the velocities are given in km/yr, we reduced our calculation results to the same dimension and rounded these to integer quantities. The changes are highlighted in blue, see lines 217, 218, 223 and 224 of our revised manuscript.

3. It is not clear from the article text, how the graphs in Fig.5 were obtained from the data shown in Fig.3? Why the line in Fig.3a is called "sinusoid" while the line in Fig.3c is treated as "appreciably different from a sinusoid"?

Following the procedure of the initial GPS time series approximation, processing of the measurement results is performed based on the standard technique for the calculation of pseudo-distances and phase measurements for each observation day (Altamimi, Z., Collilieux, X., Métivier, L., 2011. ITRF2008: an improved solution of the international terrestrial reference frame. Journal of Geodesy, 85, 457-473). The annual and semiannual variations are taken into account by the following series: $x(t)=x_0+bt+a\sin(\omega t+\varphi)+...$ (Serpelloni et al., 2013).

Because the graph shown in Fig. 3a is plotted in accordance with the above procedure, we decided to verify the extent to which it corresponds with reality. It is shown in Trofimenko et al. (2016b) that real annual site displacement curves for various regions of the world differ considerably from sinusoid and are better described by a nonlinear wave – a breather. In order to explain these dissimilarities, we added two references with brief explanations.
The changes in the text are highlighted in blue and put in the text on lines 241, 249, 250, 254, 275, 455-458 and 476-481 of the revised manuscript.

4. The "pendulum" model suggested in the paper looks very artificial. It is unclear from the text, if the authors made the calculation for such a "pendulum" oscillations of tectonic blocks or just took one of the solutions of the sine-Gordon equation. If the calculations were made – what parameters were used?

One of the objectives of the study was to find the adequate mathematical model of the vertical oscillatory movements of crustal blocks with the extraordinary trajectory pattern. In this case, the analytical solution of the sine-Gordon equation in the shape of a breather is of fundamental significance in terms of a qualitative description of recorded elementary characteristics of crustal blocks (the shape of displacement, the profile of displacement rate).
In our case, the sine-Gordon equation applied for the interacting blocks is actually postulated, but the physical interpretation of the summands of the equation is given. The validity of application of the sine-Gordon equation to the chain of blocks is proven by the fact that the implications from this equation are consistent with the results of in-situ experiment, i.e., equation (1) is the generalization of experimental data (a remarkable coincidence of solution (2) with the observation results shown in Fig. 5).
We have shown here that the sine-Gordon equation (the mathematical model, of coupled blocks-pendulums) is ***an appropriate tool*** for describing the shape of recorded displacements.

We express our sincere thanks to the anonymous reviewer for his careful reading the manuscript of this our paper as well as for familiarizing with our previous papers on the subject and making constructive comments in order to improve the quality of the manuscript.